# Factors Affecting the Quality of Sleep and Social Participation of Stroke Patients

**DOI:** 10.3390/brainsci13071068

**Published:** 2023-07-13

**Authors:** Ho-Sung Hwang, Hee Kim

**Affiliations:** Department of Occupational Therapy, Konyang University of Occupational Therapy, 158 Gwanjeodong-ro, Seo-gu, Daejeon 35365, Republic of Korea; hosung27@naver.com

**Keywords:** sleep quality, social participation, stroke, cerebral vascular accident, path analysis

## Abstract

(1) Background: Stroke patients are exposed to various psychosocial factors, such as depression, anxiety, and stress, which can cause problems with respect to their quality of sleep and social participation. (2) Objectives: We analyzed the causal relationships between the factors affecting sleep quality and social participation in hospitalized and community-based stroke patients using a path analysis model. (3) Methods: A questionnaire survey was administered to stroke patients from June to November 2020 using the Pittsburgh Sleep Quality Index to assess sleep quality; the Beck Depression Inventory to assess depression; the Beck Anxiety Inventory to assess anxiety; the Stress Scale to assess stress; and the Reintegration to Normal Living Index to assess participation. The data thus obtained were subjected to descriptive statistics, frequency analysis, and Pearson correlation analysis. In addition, anxiety, stress, and spasticity were set as exogenous variables to perform path analysis of their causal effects on depression (parameter), sleep quality, and social participation (final endogenous variables). (4) Results: In total, 145 people participated in this study, and the data of 129 people, excluding 16 insincere respondents, were analyzed. The general characteristics of the subjects comprised 84 males (65.1%) and 45 females (34.9%). Poor sleep quality comprised 54.3%, and good sleep quality comprised 45.7%, where the average age was 58.29 ± 15.46 years and the duration from onset was 39.73 ± 51.49 months. This study confirmed correlations between social participation and sleep quality, spasticity, depression, anxiety, and stress. Path analysis also revealed that anxiety and stress led to depression and that depression is a risk factor for sleep quality and social participation. (5) Conclusions: Sleep quality and social participation in stroke patients play an important role in rehabilitation. By carrying out this study, direct and indirect factors that affect sleep quality and social participation were analyzed, and the quality level in rehabilitation treatment should be improved from a structural point of view when approaching psychosocial factors.

## 1. Introduction

Stroke is a chronic disease, the worldwide incidence of which remains high due to aging populations [1,2,3]. Stroke is a central nervous system disease leading to brain tissue dysfunction due to cerebral ischemia or hemorrhage, causing sleep disorders and complex problems that affect motor and cognitive functions [4]. Sleep disorders affect 20–78% of stroke patients [5]. Considering the important role that sleep plays in learning, memorizing, and acquiring motor skills, normal sleep is crucial for post-stroke recovery [6]. Post-stroke treatment with respect to abnormal sleep patterns positively affects physical functions and psychosocial factors [7].

Previous studies on the psychosocial factors in stroke patients point to anxiety and depression as the main psychological problems where about 23% of patients with an initial stroke experience anxiety and about 19% experience depression within six months after onset [8]. Approximately 25% of patients report experiencing anxiety, and 30% of patients suffer from depression 6 months after their stroke [9]. Post-stroke psychosocial factors are directly linked to an individual’s social participation [10]. Social participation occurs when a person actively performs an occupation or participates in daily life with purpose and meaning [11]. It is an essential part of an individual’s well-being and engagement in daily living, which are important in their sociocultural context [12]. The International Classification of Functioning, Disability, and Health (ICF) defines social participation as an “involvement in a life situation” from the viewpoint of functioning [13]. Stroke patients have retracted activity and social participation and, thus, have an impaired quality of life.

As a stroke progresses from an acute to a chronic stage, psychosocial problems arise with increasing intensity [14,15]. Experiences of depression and anxiety are exacerbated and accompanied by a sleep-quality decline [16]. Stroke patients who used to live independent, healthy lives experience extreme stress in the process of relying on others’ help to perform activities of daily living (ADL) due to a decline in physical and cognitive functioning and language impairment [17]. Excessive stress acts as an obstacle to functional recovery after a stroke by affecting autonomic nervous system activity and hormone secretion and weakening the immune function [18]. Research has demonstrated that a higher stress index is associated with a lower quality of sleep [19]. Anxiety, depression, and stress ultimately deteriorate sleep quality and adversely affect social participation. Sleep quality decline in stroke patients leads to daytime drowsiness which negatively affects social participation in ADL and rehabilitation [20].

Studies have demonstrated the adverse effects of depression and anxiety on sleep quality and social participation [16,21], as well as the significant correlations between depression, anxiety, and stress in stroke patients [19,22,23]. However, little research has investigated the causal relationship between post-stroke psychosocial factors such as depression, anxiety, and stress; sleep quality; and social participation. It is also necessary to determine the extent to which spasticity, a major complaint among post-stroke physical symptoms, affects sleep quality and social participation. Thus, this study analyzed the extent to which spasticity, depression, anxiety, and stress affect sleep quality and social participation using a path analysis model.

## 2. Method

### 2.1. Research Subjects and Period

This study was approved by the Bioethics Committee (KYU-2020-069-01) at Konyang University. A survey was conducted from June to November 2020 and involved stroke patients recruited from university hospitals, rehabilitation hospitals, rehabilitation care centers, and public health centers in the Republic of Korea if they met the following criteria: (1) Patients were diagnosed with a stroke by a specialist; (2) patients who obtained a score of 24 or higher in the Korean version of the Mini-Mental State Examination (MMSE-K); (3) patients who fully understood the questionnaire and signed the informed consent form. The participants understood all explanations concerning the research purpose, procedures, data collection, and personal information protection and gave their voluntary consent to take part. Questionnaires were administered and collected by the occupational therapist in each participating facility and were completed with the aid of the occupational therapist or a caregiver. The number of samples was set using G-power 3.1.9.7. When the effect size was set to 0.15, the α error was 0.05, power was set to 0.95, and the number of predictors was set to 4; the number of samples from 129 subjects was obtained. A total of 145 patients participated. Among them, 129 questionnaires were used for the final analysis, excluding 16 where the respondent did not answer the questions, did not understand the question, or did not clearly select an answer.

### 2.2. Research Tools

#### 2.2.1. Pittsburgh Sleep Quality Index

A subjective sleep quality assessment was performed using the self-reported Pittsburgh Sleep Quality Index (PSQI) [24,25,26], which is widely used to assess sleep quality. It comprises 19 questions categorized into 7 components: sleep quality, sleep onset latency, sleep duration, sleep efficiency, sleep disturbances, use of sleeping medication, and daytime dysfunction. Each item is rated on a 4-point scale in two response option categories: sleep quality (0 = very good, 1 = slightly good, 2 = slightly bad, and 3 = very bad) and sleep onset latency, sleep duration, sleep efficiency, sleep disturbances, daytime dysfunction, and use of sleeping medication (0 = none, 1 = less than once a week, 2 = 1 or 2 times a week, and 3 = 3 or more times a week), with the total score ranging between 0 and 21 points. The total score of all components was obtained by adding up the item scores (scores exceeding 5 indicate poor sleep). Cronbach’s α for this instrument was 0.73 [24], and test–retest reliability was 0.87 [25]. In this study, Cronbach’s α was 0.70.

#### 2.2.2. Beck Depression Inventory

Depression in stroke patients was assessed using the Beck Depression Inventory (BDI) developed by Beck in 1961, and it was translated into Korean, revised, and standardized for use with stroke patients [26]. It comprises 21 items rated on a scale from 0 to 3, with the total score ranging from 0 to 63. A total score of 21 or higher indicates clinically significant depressive symptoms where the higher the total score, the more severe the depression, with 0 to 9 indicating a normal range; 10 to 15, mild depression; 16 to 23, moderate depression; and 24 to 63, extremely severe depression [27]. Cronbach’s α was 0.88, and test–retest reliability was 0.84 [28]. In this study, Cronbach’s α was 0.82.

#### 2.2.3. Beck Anxiety Inventory

Anxiety in stroke patients was assessed using the Beck Anxiety Inventory (BAI) developed by Beck et al. in 1988 and translated into Korean by Yook and Kim [29]. Each item in this 21-item instrument is rated on a 0 to 3 scale. On a total scale ranging from 0 to 63, 22 to 26 indicates mild to moderate anxiety, 27 to 31 indicates severe anxiety, and 32 or more indicates extreme anxiety. A total score of 22 was set as the cutoff for the “state of anxiety that requires observation and intervention”. Respondents rated the extent of anxiety symptoms they had experienced over the past week. Cronbach’s α was 0.915 [29]. In this study, Cronbach’s α was 0.90.

#### 2.2.4. Stress Scale

Stress experienced by stroke patients due to various restrictions was assessed using an instrument comprising 21 items that are categorized into 3 domains: personal stress (9 items), family stress (6 items), and social stress (6 items) [30,31]. Each item is rated on a 5-point Likert scale (from 5 = strongly agree to 1 = strongly disagree) with a maximum score of 105. A higher total score indicates a higher stress level. Cronbach’s α in the Korean version was 0.93 [31]. In this study, Cronbach’s α was 0.80.

#### 2.2.5. Reintegration to Normal Living Index

The Reintegration to Normal Living Index (RNLI) measures the extent to which individuals with traumatic or neurological conditions achieve reintegration into normal social participation. The questionnaire assigns a score between 0 and 10 for 11 questions [32,33,34] and measures activity and social participation among the ICF domains. As a self-reported assessment tool, the RNLI evaluates an individual’s ability to reintegrate into normal life after illness or injury, specifically their levels of community reintegration, daily activities, social participation, and personal emotional aspects [35]. Cronbach’s α was 0.87, with the coefficients between items ranging from 0.37 to 0.67 [36]. In this study, Cronbach’s α was 0.98.

### 2.3. Analysis Methods

#### 2.3.1. Path Analysis

A path analysis model is used to detect and explain the causal relationships between variables that affect the dependent variable. A range of cases can be explained in this model via the increase and decrease in variables. Using a path analysis model, a model optimized for theoretical assumptions can be developed, which enables model estimation regarding intervariable causal relationships.

#### 2.3.2. Prediction Model

Based on previous studies, a hypothesis was set up to analyze the effects on sleep quality and social participation. It was hypothesized that each exogenous variable would show a correlation with each other and affect the quality of sleep and that the quality of sleep would affect social participation. Therefore, initially, anxiety, stress, depression, and spasticity were set as exogenous variables, and sleep quality was set as an intermediate parameter to determine whether they affect social participation (Figure 1).

#### 2.3.3. Analysis Methods

The collected data were encoded and entered into SPSS 22.0 for statistical data analysis. To analyze the participants’ general characteristics, descriptive statistics and frequency analysis were performed. Pearson’s correlation analysis was performed to analyze the correlation between the raw scores of sleep quality, stiffness, depression, anxiety, stress, and social participation. Amos 20.0 was used for prediction model testing and the path analysis of the causal effects of exogenous variables (anxiety, stress, and spasticity) and depression (intermediate parameter) on sleep quality and social participation (final endogenous variables). For path analysis, the maximum likelihood (ML) method was applied with the number of bootstraps set at 2000.

## 3. Results

### 3.1. Participants’ General Characteristics

A total number of 145 people participated in this study. Among them, 129 questionnaires were used for analysis, excluding 16 in which the respondent did not answer the question, did not understand the question, or did not clearly select the answer. The general characteristics of the subjects were age, duration from onset, social participation rate, sleep quality, paralyzed side of the body, diagnosis, final education level, current job, marital status, monthly income, residential area, currently used assistance device, and spasticity level. In addition, an independent sample *t*-test was conducted to verify the homogeneity between groups according to gender. It was confirmed that they were identical except for paralyzed side of the body and education. The overall mean age was 58.29 ± 15.46 years, and the mean duration was 39.73 ± 54.49 months. Table 1 summarizes the participants’ general characteristics.

### 3.2. Correlations between Sleep Quality, Social Participation, Depression, Anxiety, Stress, and Spasticity

Depression showed a significant positive correlation with anxiety and stress and a significant negative correlation with sleep quality and social participation. Spasticity also showed a significant negative correlation with social participation. Intervariable correlations involving all variables (endogenous variables, parameters, and exogenous variables) were generated (Table 2).

### 3.3. Measurement Model and Model Fit

To identify the variables affecting the quality of sleep and social participation in stroke patients, we presented a final model for the analysis of causal relationships involving the exogenous variables (anxiety, stress, and spasticity) and the intermediate parameter: depression (Figure 2). The model fit indices in the measurement model are presented in Table 3. To apply the best hypothesis model, it is desirable to use various model fit indices [37]. Among the fit indices in the measurement model, with χ^2^ = 11.712 and *p* = 0.165 (higher than the significance level of 0.05); values of TLI (0.935), NFI (0.904), CFI (0.965), and GFI (0.971) all exceeding 0.90; and an RMSEA (0.060) lower than 0.1, the null hypothesis that there is no difference between the measurement model and the characteristics of the collected data was adopted.

Table 4 presents the results of the path coefficient analysis of the final model. In path analysis, path coefficients indicate explanatory power and predictive power [38]. The absolute values of the standardized path coefficient (≤0.10, ~0.30, and ≥0.50) indicate small, medium, and large effect sizes, respectively [39]. From this perspective, an analysis of the direct path of each pathway size can lead to the interpretation that the effect size is moderate, except for those of depression on sleep and spasticity on social participation.

### 3.4. Direct, Indirect, and Total Effects of Variables

Table 5 presents the path analysis results of the direct, indirect, and total effects of the measurement model’s variables that affect depression, quality of sleep, and social participation.

## 4. Discussion

This study presented a model for factors that affect the quality of sleep and social participation in stroke patients. Based on data from 129 stroke patients, the causal relationships between the influencing factors (depression, anxiety, stress, and spasticity) and sleep quality and social participation were examined via path analysis. The final pathway model demonstrated that the higher the anxiety and stress, the higher the depression; the higher the depression, the lower the quality of sleep and social participation. Likewise, stress and spasticity were found to adversely affect social participation in stroke patients. 

Among the stroke patients participating in this study, a higher proportion (54.3%) had poor sleep quality compared with those who had high sleep quality. According to a study conducted in China, 64.7% of stroke patients had poor sleep quality [21], and a Brazilian study noted that 70.6% of stroke patients had sleep problems [40]. A Canadian study reported the prevalence of sleep deprivation among stroke patients to be 47% [23]. The current study also demonstrated that the recognition of sleep quality problems in stroke patients is an important issue. Despite differences with respect to the national level, geographic location, ethnicity, and research methods, there is a consensus that stroke patients have poor sleep quality.

In this study, stroke patients with poor sleep quality were found to be more depressed and have more difficulty with participation compared with those with normal sleep quality. It was also found that anxiety and stress were directly associated with depression. Previous research has also demonstrated a common incidence of depression and anxiety among acute stroke patients and a correlation between the two variables [15]; moreover, a significant relationship between depression and stress was observed, which even led to interruptions in the post-stroke rehabilitation process [41]. These study results support the results in this study showing that depression induced by anxiety and stress adversely affects participation. 

The path analysis in this study revealed that depression directly affects the quality of sleep. Previous studies also reported on the correlation between sleep quality and depression and the association between depression and sleep quality, sleep onset latency, habitual sleep efficiency, and daytime dysfunction [23]. It was also reported that neurological disorders accompanying depression in stroke patients act as a factor in deteriorating their quality of sleep [20]. Short sleep duration after a stroke was reported to be a risk factor for depression [42], and the severity of sleep apnea, which is a common post-stroke symptom, was also reported to be a risk factor for depression [43]. In addition, psychosocial disorders experienced by stroke patients interfere with sleep [44]. In this study, depression was analyzed to be a direct risk factor affecting the quality of sleep, which is consistent with the results in previous studies reporting that the psychosocial disorders experienced by stroke patients interfere with sleep [45]. That is, anxiety and stress, which are psychosocial factors in stroke patients, affect depression, and depression affects sleep quality. 

The analysis in this study identified the increase in depression and stress as the direct risk factor in diminished participation. Depression in stroke patients is a main symptom throughout the acute and subacute phases in stroke patients [46]. While stroke patients usually focus on improving physical functions in the initial stage of treatment and are actively engaged in rehabilitation procedures, depression in the early stages can become an obstacle to rehabilitation intention and functional recovery [47]. In addition, stress-induced fatigue has a negative effect on physical ADL and participation in rehabilitation [48]. With regard to life satisfaction, depression was found to hinder satisfaction [49], and mental stability and participation were identified as important factors for life satisfaction [50]. Therefore, the promotion of post-stroke participation in daily life should be preceded by intervention strategies to reduce depression and stress.

In stroke patients, spasticity has a negative effect on physical function, participation, and quality of life [51]. In particular, upper extremity spasticity leads to upper extremity dysfunction with prolonged joint contracture due to pain and reduced upper extremity motion resulting from fibrosis in joints, muscles, and nerves due to muscle weakness on the paretic side [52]. The upper extremity function in stroke patients is one of the most important functions for performing ADL [53]. As a basic function for specific movements, such as dressing, writing, and eating, upper extremity function is an essential factor for performing ADL [54]. Therefore, therapeutic strategies to counter spasticity, which causes deterioration in upper extremity function, can be viewed as an essential factor in the rehabilitation of stroke patients.

In this study’s planning phase, an analysis of previous studies led to the prediction that there would be a direct or indirect path between sleep quality and social participation [8,9,16,21]; however, no significant path was found. When the model fit in the initial prediction model (Figure 1) was checked, TLI (−0.540), NFI (0.606), CFI (0.589), and GFI (0.900) were all less than 0.90, and RMSEA was 0.306 (more than 0.1), so the fit in the model was not established. To establish the fit in the model, the path was rearranged based on the correlation analysis results. The model fit in the reset final model was measured because TLI (0.935), NFI (0.904), CFI (0.965), and GFI (0.971) were all over 0.90, and RMSEA was 0.060, which was less than 0.1. The fit in the model was established. In contrast to previous findings reporting that a significant correlation exists between sleep quality and social participation [55], no direct or indirect association was observed between the two factors in our final path analysis model. The analysis in this study revealed significant differences in depression and participation between the sleep disorder and normal-sleep groups and a significant correlation between sleep quality and participation. When looking at the difference between the groups in the sleep disorder and normal groups, significant differences were confirmed with respect to depression and participation, and a significant relationship was also confirmed in the correlation between sleep quality and participation. From this, it was assumed that sleep quality and the participation of stroke patients will have direct and indirect pathways between each other. However, no pathway was found between the two variables. Sleep and participation fall under different categories in the ICF classification. In the first classification of ICF, sleep is classified as a mental function among body functions, and participation in activities falls under different categories, such as body function, body structure, activities and participation, and environmental factors. It cannot be ruled out that these different classification categories may influence the association between the two variables.

Although many studies have analyzed the correlations between depression and anxiety, stress, spasticity, sleep quality, and social participation in stroke patients [19,22,23,56], few have analyzed the direct and indirect pathways between individual variables. Thus, this study has clinical significance due to its analysis of direct and indirect effects via the path analysis of individual factors. However, the study has some limitations. First, in addition to the physical and mental factors that we examined, economic factors, education level, region of residence, and the use of assistive devices can influence sleep quality and social participation. Second, when considering the characteristics of the study’s subjects, there is a possibility that a relatively large number of males, high levels of education, many married people, many people living in cities, and many users of manual wheelchairs will result in a biased analysis. Third, the average duration from onset after stroke was 3 years, and there were limitations in that the standard deviation was large and the age of the subjects was relatively young. Fourth, since this study is a cross-sectional study, it is unreasonable to generalize its research results, and there is the limitation that this is not a controlled study design. Addressing these limitations will require the use of larger sample sizes and the analysis of various physical and psychological factors using controlled study designs. In addition, since psychosocial factors related to sleep quality and social participation were analyzed, intervention-based experimental research should be conducted based on the results of this study.

## 5. Conclusions

This study carried out path analysis to analyze the causal relationships between depression, anxiety, stress, and spasticity that affect sleep quality and social participation in stroke patients. Depression in stroke patients can be viewed as a risk factor for anxiety and stress, which negatively affects sleep quality and social participation. The overall fit in the pathway model for sleep quality and social participation was good. This study is clinically significant in that it tested the factors directly or indirectly affecting sleep quality and social participation in stroke patients and explained their causal relationships. A clear causal relationship was established in the association between depression and sleep quality. Various approaches are required for the post-stroke rehabilitation process in the future, with preceding interventions for psychological–emotional disorders in addition to physical rehabilitation.

## Figures and Tables

**Figure 1 brainsci-13-01068-f001:**
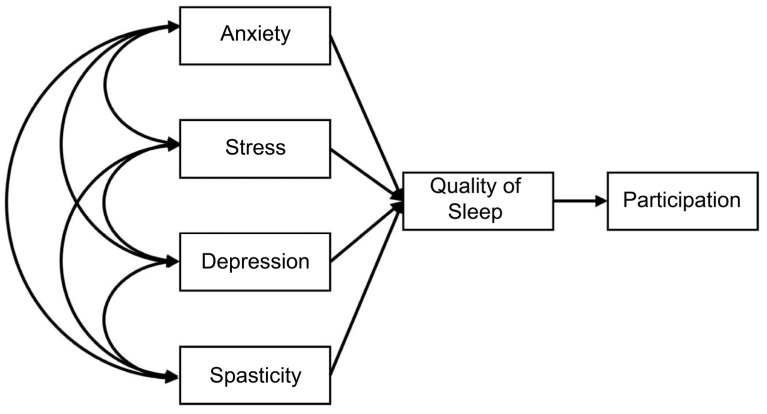
Prediction model.

**Figure 2 brainsci-13-01068-f002:**
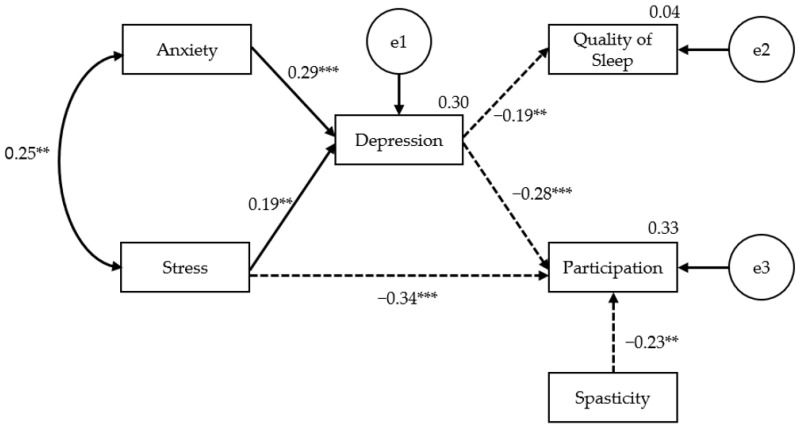
The final path model for quality of sleep and participation. ** *p* < 0.01; *** *p* < 0.001. → Positive(+) effect, 
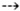
 negative(−) effect.

**Table 1 brainsci-13-01068-t001:** General characteristics of subjects (N = 129).

Characteristics		N (%)	Male (n = 84)	Female (n = 45)	*t*
			n (%)	n (%)	
Social participation rate	90–100%	16 (12.4)	8 (6.2)	8 (6.2)	0.375
	80–89%	8 (6.2)	8 (6.2)	0 (0.0)	
	70–79%	9 (7.0)	7 (5.4)	2 (1.6)	
	60–69%	18 (14.0)	11 (8.6)	7 (5.4)	
	50–59%	14 (10.8)	10 (7.7)	4 (3.1)	
	40–49%	12 (9.3)	8 (6.2)	4 (3.1)	
	30–39%	12 (9.3)	5 (3.9)	7 (5.4)	
	20–29%	13 (10.1)	11 (8.6)	2 (1.5)	
	10–19%	14 (10.8)	9 (6.9)	5 (3.9)	
	0–9%	13 (10.1)	7 (5.4)	6 (4.7)	
Sleep quality	Poor	70 (54.3)	44 (34.1)	26 (20.2)	0.583
	High	59 (45.7)	40 (31.0)	19 (14.7)	
Paralyzed side of the body	Rt.	60 (46.5)	48 (37.2)	12 (9.3)	2.644 **
	Lt.	55 (42.7)	29 (22.5)	26 (20.2)	
	Both	3 (2.3)	1 (0.8)	2 (1.5)	
	Normal	11 (8.5)	6 (4.6)	5 (3.9)	
Diagnosis	Infarction	70 (54.3)	43 (33.4)	27 (20.9)	0.953
	Hemorrhage	59 (45.7)	41 (31.7)	18 (14.0)	
Education	≥University	42 (32.5)	27 (20.9)	15 (11.6)	2.144 *
	High school	46 (35.7)	36 (27.9)	10 (7.8)	
	Middle school	15 (11.6)	10 (7.7)	5 (3.9)	
	Elementary School	21 (16.3)	10 (7.7)	11 (8.5)	
	<Elementary School	5 (3.9)	1 (0.8)	4 (3.1)	
Current Job	Employee	16 (12.4)	12 (9.3)	4 (3.1)	0.182
	Self-employed	28 (21.7)	20 (15.5)	8 (6.2)	
	Student	2 (1.6)	0 (0.0)	2 (1.6)	
	Housewife	14 (10.9)	0 (0.0)	14 (10.9)	
	Retirement	7 (5.4)	7 (5.4)	0 (0.0)	
	Retirement (health problem)	57 (44.1)	41 (31.7)	16 (12.4)	
	Retirement (other reason)	4 (3.9)	4 (3.1)	1 (0.8)	
Marital status	Married	79 (61.2)	56 (43.4)	23 (17.8)	2.695 **
	Single	21 (16.3)	15 (11.6)	6 (4.7)	
	Divorce	9 (7.0)	5 (3.9)	4 (3.1)	
	Bereaved	16 (12.4)	7 (5.4)	9 (7.0)	
	Separation	4 (3.1)	1 (0.8)	3 (2.3)	
Monthly income (KRW)	>3,000,000 (>USD 2295)	23 (17.8)	15 (11.6)	8 (6.2)	0.056
	2,000,000–3,000,000 (USD 1530–USD 2295)	30 (23.3)	21 (16.3)	9 (7.0)	
	1,500,000–2,000,000 (USD 1147–USD 1530)	26 (20.2)	14 (10.9)	12 (9.3)	
	1,000,000–1,500,000 (USD 765–USD 1147)	17 (13.2)	12 (9.3)	5 (3.9)	
	<1,000,000 (<USD 765)	33 (25.6)	22 (17.1)	11 (8.5)	
Residential area	Metropolis	81 (62.8)	55 (42.6)	26 (20.2)	0.439
	Small-medium cities	24 (18.6)	13 (10.1)	11 (8.5)	
	Small town	24 (18.6)	16 (12.4)	8 (6.2)	
Assistance device	None	54 (41.8)	34 (26.4)	20 (15.4)	0.629
	Manual wheelchair	44 (34.1)	30 (23.2)	14 (10.9)	
	Cane	21 (16.3)	13 (10.1)	8 (6.2)	
	Electric wheelchair	5 (3.9)	4 (3.1)	1 (0.8)	
	Walker	5 (3.9)	3 (2.3)	2 (1.6)	
Spasticity	MAS 0	43 (33.3)	27 (20.9)	16 (12.4)	1.340
	MAS 1	31 (24.0)	16 (12.4)	15 (11.6)	
	MAS 1+	24 (18.6)	18 (14.0)	6 (4.7)	
	MAS 2	25 (19.4)	19 (14.7)	6 (4.7)	
	MAS 3	6 (4.7)	4 (3.1)	2 (1.6)	
Age (year. M ± SD)		58.29 ± 15.46			0.252
Duration from onset (months, M ± SD)		39.73 ± 51.49			0.690

* *p* < 0.05; ** *p* < 0.01. MAS: Modified Ashworth Scale; KRW: Korean won.

**Table 2 brainsci-13-01068-t002:** Inter-correlations of variables (N = 129).

	Quality of Sleep	Spasticity	Depression	Anxiety	Stress	Participation
Quality of sleep	1					
Spasticity	0.094 **	1				
Depression	−0.195 **	0.051 **	1			
Anxiety	−0.145 **	0.121 **	0.393 **	1		
Stress	−0.118 **	−0.114 **	0.473 **	0.252 **	1	
Participation	0.178 *	−0.203 **	−0.456 **	−0.314 **	−0.446 **	1

* *p* < 0.05; ** *p* < 0.01.

**Table 3 brainsci-13-01068-t003:** Test of model fit.

χ^2^ (*p*)	df	TLI	NFI	CFI	GFI	RMSEA
11.712 (0.165)	8	0.935	0.904	0.965	0.971	0.060

TLI: Tucker–Lewis index; NFI: normed fix index; CFI: comparative fit index; GFI: goodness of fit index; RMSEA: root mean square error of approximation.

**Table 4 brainsci-13-01068-t004:** Path coefficients of the model.

			β	S.E.	C.R.
Depression	←	Stress	0.399	0.005	5.239 ***
Depression	←	Anxiety	0.292	0.112	3.837 ***
Sleep	←	Depression	−0.195	0.036	−2.246 *
Participation	←	Depression	−0.283	2.018	−3.456 ***
Participation	←	Spasticity	−0.225	1.696	−3.122 **
Participation	←	Stress	−0.336	0.135	−4.099 ***

* *p* < 0.05; ** *p* < 0.01; *** *p* < 0.001. S.E. = Standard error; C.R. = critical ratio.

**Table 5 brainsci-13-01068-t005:** Path analysis of standardized direct, indirect, and total effects.

Variable		Spasticity	Stress	Anxiety	Depression
Depression	Indirect Effects				
	Direct Effects		0.399 **	0.292 **	
	Total Effects		0.399 **	0.292 **	
Participation	Indirect Effects		−0.113 **	−0.083 **	
	Direct Effects	−0.225 **	−0.336 **		−0.283 **
	Total Effects	−0.225 **	−0.449 **	−0.083 **	−0.283 **
Quality of Sleep	Indirect Effects		−0.078 *	−0.057 *	
	Direct Effects				−0.195 *
	Total Effects		−0.078 *	−0.057 *	−0.195 *

* *p* < 0.05; ** *p* < 0.01.

## Data Availability

Not applicable.

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
