# Peer review of "Factors Affecting the Quality of Sleep and Social Participation of Stroke Patients"

_brainsci, 2023, doi:10.3390/brainsci13071068_

Round 1

Reviewer 1 Report

Comments and Suggestions for Authors

Thank you for submitting your manuscript. 

Having carefully reviewed your work, I would like to provide you with some feedback and suggestions for improvement. Firstly, it is essential to recognize that the utilization of path analysis within a cross-sectional design does not permit the assumption of causal relationships, as you have claimed in the discussion and conclusion sections. Causal inferences generally require a longitudinal or experimental design to establish a stronger basis.

Additionally, upon examining the correlations presented in Table 2, it is evident that the majority of them are negligible to fair in strength. Consequently, relying on these modest results alone would not be sufficient to conclude a strong relationship between the variables under investigation.

To enhance the scientific rigor and validity of your study, I recommend considering the following suggestions:

 1. Revisit the discussion and conclusion sections to reframe the language used when discussing the causal relationships. Emphasize the limitations of inferring causality based on a cross-sectional design.

 2. Provide a more nuanced interpretation of the correlations presented in Table 2, considering their modest strength. Explore potential explanations for these findings,.

 3. Consider alternative research designs that could help establish a stronger basis for causal claims. Longitudinal studies or experiments with controlled variables could offer valuable insights into the relationships between the variables you investigated.

I also added the following minor comments for you.

Abstract:

Line 14-15: “Beck Depression Inventory to assess anxiety”. Correct this sentence to the following: Beck Depression Inventory to assess depression, Beck anxiety Inventory to assess anxiety.

Introduction:

Line 55: “social participation: Sleep quality decline”: change the “:” to “.”

Methods:

Lines 154-155. Correct the following sentence to reflect Figure 1: “analysis of the causal effects of the exogenous variables (anxiety, stress, and spasticity) and depression (intermediate parameter) on sleep quality…”

Lines 160-161. “The overall mean age was 58.29±15.46 years, and the mean duration was 39.73±54.49 years”: change years into months and clarify what duration you mean.

Results:

Table 2: Write variable names on the top of each column.

Lines 166-172: Delete this because it is reiteration for Table 2.

Line 198: change “quality of life” to “quality of sleep”

Line 199-210: Delete this because it is reiteration for Table 5.

Discussion:

Line 215: change “…stiffness)” to “…spasticity)”.

Line 228: “In this study, however, depression was analyzed to be a direct cause affecting the quality of sleep…” this statement is not true.

Comments on the Quality of English Language

Minor editing of English language required.

Author Response

Thank you for commenting on our research. We will upload the modified file.

Reviewer 2 Report

Comments and Suggestions for Authors

This paper explored psychosocial factors (anxiety, depression, stress) and spasticity that affect sleep quality and social participation in stroke patients. The method of path analysis is used to solve the problem of causality among variables. On the whole, the content of this paper is complete and has certain practical significance. I have the following comments and questions to discuss with the authors.

Comment 1: The conclusion part of the abstract describes the significance of the research, rather than the conclusion of the research. So it is suggested to modify it as the conclusion of this study.

Comment 2: The overall hypothesis proposed in the introduction lacks relevant research evidence and theoretical basis, mainly including: the relationship and difference between stress, anxiety and depression; The relationship between sleep quality and social participation; The reason for the spasticity as another variable.

Comment 3: The model reported in the results section is completely inconsistent with the original assumptions. In the hypothesis, anxiety, depression, stress, and spasticity are exogenous variables, sleep quality is the mediating variable, and social participation is the dependent variable. However, in the outcome, only anxiety and stress are exogenous variables, and depression is the mediating variable of sleep quality and social participation. Although the situation is briefly explained in the discussion, I believe that the model fitting results that are consistent with the hypothesis should be reported in the results, and the reasons why the final fitting results are not consistent with the hypothesis should be detailed in the discussion.

Comment 4: Why treated all variables in the study as explicit variables using path analysis instead of structural equation models?

Comment 5: Can stroke patients accurately recall their sleep in the nearly month? How is the Pittsburgh Sleep Quality Index Scale validity in stroke patients?

Comment 6: The three-line table is not standardized, and there should be no solid line in the table. All abbreviations in table1 need to be explained below the table.

Comment 7: The discussion section is not in-depth and does not extend the results properly, including the failure to explain the influence of anxiety and stress on depression; The reasons for the poor fit of the assumed model are not explained in detail.

Comment 8: Limitations and prospects, the overall feeling is relatively empty. There is no discussion of the real possible flaws of this paper or of future topics worthy of study. The demographic variables mentioned herein can be calculated as covariates in this study.

Author Response

(The authors gave the same response as above.)

Reviewer 3 Report

Comments and Suggestions for Authors

Factors Affecting Quality of Sleep and Social Participation in Stroke Patients is an interesting research topic.

The following items are well described in this study

∙ Adequate number of subjects

∙ Approval from an appropriate ethical review board

∙ Purposeful path model analysis

This study is expected to provide new suggestions for improving the quality of life of stroke patients.

Comments on the Quality of English Language

No particular English grammar problems were identified.

Author Response

Thank you for your interest in our research.

Reviewer 4 Report

Comments and Suggestions for Authors

This paper report on a multi-centre study of factors affecting sleep quality and social participation among stroke patients. The authors found correlations between social participation and sleep quality, spasticity, depression, anxiety, and stress. Path analysis revealed that anxiety and stress lead to depression and that depression is a risk factor for sleep quality and social participation.

The authors may wish to address the following issues:

1. Abstract – Results – to add demographic data; to add relevant statistics
2. Methods - lines 82-85 - ‘A total of 145 patients … did not clearly select an answer’ should be in the Results
3.  Fig 1 - by the arrows, are the authors suggesting that anxiety, stress and depression lead to spasticity?
4. Results – please show the distribution of the data of the end-points ie sleep quality and social participation, if it is possible in a simple way. Why were the baseline characteristics not included in the analysis (also mentioned by the authors in lines 272-274?
5. Table 1 – I suggest the authors use a consistent hierarchy on responses within each variable. ‘Affected side’ – of the body or brain? Are ‘job’, ‘marital status’, ‘income’ at the time of stroke or at the time of the study?
6. Discussion – lines 269-272 – there is a lot of overlap between these 2 sentences – it is in essence just 1 grouping of limitations. Additional limitations may be the population is relatively young, well-educated, more male, more urban, and assessed at 3 years after stroke with a wide SD….

Author Response

(The authors gave the same response as above.)

Round 2

Reviewer 1 Report

Comments and Suggestions for Authors

Thank you for yor positive response to review points.

My only concern is with using the word "cause" in the discussion and conclusion sections. Find an alternative and appropriate word such as "risk factor" or "potential predictor" since your research design is cross sectional.

Author Response

Thank you for your nice review on our article. I have corrected and supplemented what you suggested.

Reviewer 4 Report

Comments and Suggestions for Authors

This paper is a revised submission of a report on a multi-centre study of factors affecting sleep quality and social participation among stroke patients.

The authors have addressed some of my concerns. The remaining ones are:

1. Abstract – Results – to add demographic data quantitatively eg 145 subjects, mean age…, SD..,  …. % female. etc; To add relevant statistics as I requested initially to support the claim of correlations
2. Methods – initial lines 82-85 - ‘A total of 145 patients … did not clearly select an answer’ needn’t be deleted completely, but instead inserted into the first paragraph of Results
3. Results – I think it is appropriate to show the distribution of the data of the end-points ie sleep quality and social participation, if it is possible in a simple way. I think it important that the baseline characteristics be included in the analysis
4. Table 1 – for a consistent hierarchy on responses within each variable, I suggest the authors reverse the sequence for education, assistance device
5. Discussion –Additional limitations may be the population is relatively young, assessed at 3 years after stroke with a wide SD, as I had mentioned previously

Author Response

(The authors gave the same response as above.)
